# Effects of Fatigue on Ankle Flexor Activity and Ground Reaction Forces in Elite Table Tennis Players

**DOI:** 10.3390/s24206521

**Published:** 2024-10-10

**Authors:** Yunfei Lu, Jun Wang, Yuanshi Ren, Jie Ren

**Affiliations:** 1China Table Tennis College, Shanghai University of Sport, Shanghai 200438, China; 2021111016@sus.edu.cn (Y.L.); renyuanshi1993@163.com (Y.R.); 2Shanghai Key Laboratory of Intelligent Manufacturing and Robotics, Shanghai University, Shanghai 200444, China; jun.wang@grenoble-inp.fr

**Keywords:** muscle fatigue, table tennis, ankle contraction, electromyography

## Abstract

Fatigue specifically affects the force production capacity of the working muscle, leading to a decline in athletes’ performance. This study investigated the impact of fatigue on ankle flexor muscle activity and ground reaction forces (GRFs) in elite table tennis players, with a focus on the implications for performance and injury risk. Twelve elite male table tennis athletes participated in this study, undergoing a fatigue protocol that simulated intense gameplay conditions. Muscle activity of the soleus (SOL) and gastrocnemius lateralis (GL) muscles, heel height, and GRFs were measured using a combination of wireless electromyography (EMG), motion capture, and force plate systems. Results showed a significant decrease in muscle activity in both legs post-fatigue, with a more pronounced decline in the right leg. This decrease in muscle activity negatively affected ankle joint flexibility, limiting heel lift-off. Interestingly, the maximal anteroposterior GRF generated by the left leg increased in the post-fatigue phase, suggesting the use of compensatory strategies to maintain balance and performance. These findings underscore the importance of managing fatigue, addressing muscle imbalances, and improving ankle flexibility and strength to optimize performance and reduce the risk of injuries.

## 1. Introduction

Table tennis, categorized as a reaction sport, exemplifies this relationship due to its fast-paced nature, requiring players to react within fractions of a second to the ball’s speed and trajectory [1]. The high demands placed on reflexes and motor coordination in table tennis necessitate rigorous and repetitive training routines. These routines, while essential for developing the necessary reflexes and technical skills, also increase the risk of overuse injuries among athletes [2].

Overuse injuries in table tennis primarily arise from the repetitive stress imposed on specific muscle groups and joints during training and gameplay. These injuries commonly affect the upper limb and often occur during the training period. A study by Kondrič et al. [3], which examined injuries in top Slovenian racket sport players, found that the shoulder was the most common injured area among top table tennis players, with approximately 20% of reported injuries occurring in the shoulder. This is particularly relevant as the shoulder joint is heavily involved in the forehand pull technique, where its kinematics and dynamics significantly impact the quality of the shot [4]. High-level athletes tend to exhibit a greater range of motion in the shoulder joint during the forehand pull, along with increased internal rotation torque and angular velocity [5]. However, increasing the range of motion in the shoulder joint can place significant stress on the surrounding ligaments and muscles, potentially leading to injuries such as dominant shoulder impingement syndrome [6]. Prolonged training sessions may selectively strengthen the muscles located in front of the shoulder joint, including the subscapularis, pectoralis major, and deltoid, contributing to the risk of injury [7].

In fact, fatigue, a common consequence of intense or prolonged physical activity, plays a significant role in the onset of these injuries. When athletes are fatigued, their ability to stabilize joints and control movements effectively diminishes, resulting in increased stress on connective tissues and a higher likelihood of injury. Fatigue distinguishes between mental fatigue and muscle fatigue based on the source of exertion. Mental fatigue refers to fatigue resulting from cognitive or mental exertion [8], while muscle fatigue refers to fatigue resulting from physical exertion in the muscles [9]. Both types of fatigue can lead to subjective feelings of tiredness, lack of energy, and an increased perception of effort during subsequent physical tasks. Studies have shown that both mental and muscle fatigue can impact performance and affect the perception of effort during physical activities [2]. However, there is a key difference: muscle fatigue specifically affects the force production capacity of the working muscles, leading to a decline in physical performance, a decrease not observed with mental fatigue [10,11]. On the other hand, mental fatigue has been shown to impact skilled performance in table tennis, particularly affecting inhibitory stimuli during visuomotor tasks, suggesting that mental fatigue can interfere with the cognitive processes necessary for optimal performance [12]. Despite the recognition of muscle fatigue as a critical factor in sports-related injuries, there is a relative scarcity of research specifically addressing its effects in table tennis. A study by Aune et al. suggested that muscle fatigue negatively impacts table tennis performance, although specific details regarding the muscles involved were not provided [13]. Another study by Le Mansec et al. examined the effects of both mental and muscle fatigue on table tennis performance, demonstrating that both types of fatigue can decrease movement speed and have detrimental effects on manual dexterity and anticipation timing. Their investigation focused on muscle fatigue in the elbow flexors (biceps) and knee extensors (quadriceps) [2].

While studies have examined the impact of fatigue on specific muscle groups in table tennis, those related to plantarflexion have not been extensively explored. Given the importance of lower limb stability and footwork in table tennis, understanding how fatigue affects the plantarflexion—muscles critical for movements such as plantar flexion and dorsiflexion—is vital. When athletes experience fatigue, the muscle activity of the plantarflexion, specifically the soleus (SOL) and gastrocnemius (GL) muscles, may negatively impact their ability to achieve complete heel lift-off during movements. The SOL plays a crucial role in plantar flexion, the downward pointing motion of the foot and toes. The GL, a two-joint muscle that spans both the knee and ankle, is also primarily responsible for plantar flexion. Together, the SOL and GL generate the force required to push off the ground and propel the body forward during activities such as walking, running, and jumping [14,15,16]. When fatigued, athletes may struggle to generate sufficient force to lift the heel off the ground, leading to reduced ankle dorsiflexion and limited foot clearance. This can increase the contact area between the plantar surface of the foot and the ground, potentially resulting in a more pronounced varus or valgus angle of the ankle joint. In addition, Boozari et al. [17] found a decrease in vertical GRF after fatigue in individuals with flat feet, while Gerlach et al. [18] reported that fatigue-induced changes in running style also led to decreased GRF. These findings suggest that fatigue may negatively affect GRF.

Given the importance of this topic, this study aims to address the gap by investigating the effects of fatigue on the muscle activity of plantarflexion and its subsequent impact on ground reaction forces (GRFs) in table tennis players. We hypothesize the following:(1)There will be a reduction in ankle flexor activity and a decline in heel height from the ground following fatigue in athletes.(2)Reduced plantarflexion due to fatigue will negatively affect the generation of ground reaction forces (GRFs).

To test this hypothesis, this study used a motion capture system, a wireless electromyography (EMG) system, and a force plate system to assess muscle activity and ground reaction forces before and after inducing fatigue in the plantarflexion. A fatigue protocol was designed, and the ratings of perceived exertion (RPE) were used to measure the degree of fatigue. The collected data was then analyzed to determine whether significant differences in muscle activation patterns and ground reaction forces are evident after fatigue, thereby supporting the hypothesis that fatigue influences these variables in table tennis players. The findings are expected to contribute to a more comprehensive understanding of how muscle fatigue influences performance and injury risk in table tennis. This knowledge can inform the development of targeted training programs and injury prevention strategies, ultimately enhancing athlete performance while minimizing the likelihood of fatigue-related injuries.

## 2. Materials and Methods

### 2.1. Subjects

Twelve male undergraduate students (age 19.1 ± 1.6 years; body mass 66.2 ± 5.8 kg; height 174.0 ± 5.8 cm; training experience 11.9 ± 2.0 years) from the China Table Tennis College (CTTC) at Shanghai University of Sports volunteered to participate in this study. All participants were right-handed, elite athletes at China National Level I. They were in good physical condition and had no history of upper or lower limb injuries or diseases in the past six months. This study was approved by the Human Ethics Committee of Shanghai University of Sports. Written informed consent was obtained from all participants prior to their involvement in this study.

### 2.2. Measurement Items

#### 2.2.1. Kinematics

The kinematic data were captured using an infrared motion capture system (MC2000, ChingMu Inc., Shanghai, China), as shown in Figure 1. This system, consisting of 20 infrared cameras, is capable of providing real-time optical data and is widely used for motion capture and analysis. The system operates at an acquisition frequency of 100 Hz. A total of 15 reflective markers, each with a 14 mm diameter, were attached to the feet (Figure 2a) and the racket (Figure 2b).

#### 2.2.2. Electromyography

Electromyography (EMG) activity was recorded at 1 kHz using a wireless system (MEQ185, Trigno Wireless EMG, DELSYS Inc., Natick, MA, USA). Following skin preparation, including shaving and cleaning, in accordance with SENIAM (Surface EMG for Non-invasive Assessment of Muscles) guidelines [19], four electrodes were placed on the soleus (SOL) and the lateral gastrocnemius (GL) muscles of both legs (Figure 3), which are representative of the plantar flexor muscles [20]. The electrodes were secured with adhesive tape and remained in place throughout the experiment. All EMG signals were amplified (×500) and filtered using a band-pass filter set between 20 Hz and 400 Hz.

#### 2.2.3. Force

Plantar force data were recorded using six Kistler force plates (Model 926088, Kistler Instruments Ltd., Winterthur, Switzerland) with a sampling frequency of 1000 Hz. These multicomponent force plates provide dynamic measurements of the three orthogonal components of force ($F_{x}$, $F_{y}$, $F_{z}$) acting from any direction on the top plate.

#### 2.2.4. Vertical Jump Touch Device and Serving Robot

A jump device with a touch screen was used to measure the height of an athlete’s vertical and standing jumps, providing precise data on vertical jump performance. Additionally, a serving robot (PONGBOT M-ONE, Future-mind Inc., Shanghai, China) was employed in this study [21]. The PONGBOT M-ONE allowed for the flexible modification of key parameters such as ball speed (4–12 m/s), spin (0–80 rotations per second, including topspin and backspin), frequency (30–85 times per minute), and placement (any location on the table). This level of control created a controlled and adaptable training environment, facilitating a meticulous analysis of players’ reactions and enabling focused exploration of specific performance elements.

### 2.3. Experimental Design

All tests were performed at the CTTC training gymnasium. Before data collection, participants completed a warm-up and were briefed on the experimental procedure (Figure 4). Detailed information is provided below.

### 2.4. Experimental Procedure

#### 2.4.1. Warm-Up and Vertical Jump Test

Prior to the MVC, participants completed a 10 min warm-up session mirroring their regular training routine. The warm-up consisted of jogging, sprinting, and two-point movement exercises, which are commonly employed in table tennis training. Thereafter, the participants were instructed to perform a vertical jump from a stationary position, attempting to touch the highest point they could reach. This test was conducted three times, with the highest jump height recorded as the participant’s maximum vertical jump height (MH).

#### 2.4.2. Maximal Voluntary Contraction (MVC) Test

Participants were seated on a firm training mat with their feet fully pressed against the wall to maintain a relaxed and neutral position, ensuring the ankle joint was positioned at approximately a 90° angle [22]. The MVC test involved isometric contractions, ensuring that no joint movement occurred. During the test, participants were instructed to exert ‘as much force as possible’ [23] by performing a plantar flexion movement, engaging the posterior muscles of the lower leg, specifically the soleus (SOL) and gastrocnemius lateralis (GL). An assistant was present to ensure that the participants’ backs remained stationary and their knee joints stayed motionless, as shown in Figure 5. Each participant completed three sets of 5 s MVCs, with a 10 s rest interval between each set. None of the participants reported any pain or discomfort that would interfere with force production. The maximal EMG activity recorded for each muscle was selected to normalize the EMG signals throughout the entire fatigue task.

#### 2.4.3. Fatigue Task

After the MVCs, participants were given a 10 min rest period. The fatigue task in this study followed the protocols performed in previous research [24,25]. Participants were required to perform a series of 20 forehand strokes, followed by three sets of vertical jumps (Figure 6). If the average height of the jumps exceeded 70% of the participant’s maximal vertical jump height (MH), they were instructed to proceed with another set of 20 forehand strokes.

This sequence was repeated until the vertical jump height dropped below 70% of the MH. The data collected during the initial set of 20 forehand strokes were considered the pre-fatigue data, while the final set of 20 forehand strokes represented the post-fatigue data. The experiment was conducted to assess the impact of muscle fatigue on ankle flexion and ground reaction forces. Upon completion of the fatigue task, participants were asked to provide ratings of perceived exertion (RPE, 6–20 scales) to assess the level of effort associated with the task they had just performed [26].

### 2.5. Data Processing

#### 2.5.1. Coordinate Systems

In this study, a coordinate system was employed to represent the data acquired from the Kistler force plates and ChingMu motion capture system. The *x*-axis was defined as the line directed towards the front of the participant, with the positive direction extending from the participant towards the force plate. The *y*-axis was identified as the line intersecting the midline of the force plate, with its positive direction indicating the participant’s right side. Finally, the *z*-axis was defined as the direction perpendicular to the *x*–*y* plane, with the positive direction pointing upward. This coordinate system provided a consistent frame of reference for analyzing the data collected from both the force plates and motion capture system.

#### 2.5.2. Motion Phase Definition

As shown in Figure 7, the motion phases were defined following the methodology described in a previous study [27]. This study focused particularly on analyzing the participants’ performance during the forward phase (FP). The initiation of the FP (Figure 7a) was defined as the moment when the player begins to move the racket forward after positioning themselves to hit the ball, and the end of the FP (Figure 7b) was when the racket reaches its furthest point forward. This phase was studied to investigate alterations in the biomechanics of the lower limbs throughout the swing phase.

#### 2.5.3. Heel Height

The motion capture system was used to record the movement of participants’ feet during the forward phase (FP). The *z*-axis of the coordinate system represented the height dimension. To assess changes in heel height, the difference between the heel height during the FP and the heel height in a static position (Figure 7a) was measured for both pre- and post-fatigue phases. All data were normalized to a 100% motion cycle. The mean and standard error of the mean (SEM) were then calculated from the 20 recorded height values during the pre- and post-fatigue tasks.

#### 2.5.4. Muscle Activity Level

The electromyography (EMG) signals recorded during the pre- and post-fatigue tasks were amplified (×500) and filtered with a band-pass filter with a range of 20 Hz to 400 Hz [28,29]. Subsequently, the filtered EMG data were normalized by dividing the maximum EMG value obtained during the MVC test. The mean values were then calculated based on the maximum normalized EMG data obtained during the pre- and post-fatigue tasks.

#### 2.5.5. Maximal Ground Reaction Force (GRF)

The GRFs were carefully measured and analyzed along three axes: the *x*-axis, *y*-axis, and *z*-axis. To ensure comparability, the GRFs were normalized by dividing them by each participant’s body weight. The maximum values of these normalized GRFs were then calculated.

### 2.6. Statistical Analysis

Statistical analyses were performed using GraphPad Prism 8 (GraphPad Holdings, LLC, Solana Beach, CA, USA). Descriptive statistics were reported as means and standard errors of the mean (SEM). Normality of the data was assessed using the *Kolmogorov–Smirnov* test. Paired *t*-tests were employed to evaluate differences in maximal EMG levels and maximal ground reaction forces (GRFs) between the pre- and post-fatigue phases. The significance level for all tests was set at *p* < 0.05.

## 3. Results

All data passed the *Kolmogorov–Smirnov* test for normal distribution. Participants reported their perceived fatigue level using the RPE scale, with an average rating of 17 ± 0.98, confirming that the fatigue effect was successfully achieved. Specifically, one participant rated their fatigue at 16, five participants rated it at 17, and three participants rated it at 18.

### 3.1. Muscle Activity Levels

Figure 8 presents the muscle activity levels for each muscle during the pre- and post-fatigue phases. The results show that both the soleus (SOL) and gastrocnemius (GL) muscles exhibited significantly lower activity during the post-fatigue phase compared to the pre-fatigue phase, regardless of whether the right or left leg was measured (all *p* values < 0.001). Additionally, the decrease in electromyography (EMG) activity was more pronounced in the SOL and GL muscles of the right leg compared to the left leg.

### 3.2. Heel Height

Figure 9 depicts the trend of the heel height for each foot during the pre- and post-fatigue phases. The results show a consistent pattern for the left heel, with a decrease followed by an increase in height, reaching the lowest point at approximately 50% of the motion cycle. No significant difference in height was observed between the pre- and post-fatigue phases for the left heel. In contrast, the right heel exhibited an initial increase in height followed by a decrease, peaking around 50% of the motion cycle. Notably, Figure 9b shows a significant reduction in right heel height during the post-fatigue phase compared to the pre-fatigue phase, particularly from 20% to the end of the motion cycle.

### 3.3. Maximal Ground Reaction Force (GRF)

Table 1 displays the maximal GRFs in three vectors during the pre- and post-fatigue phases. Notably, what stands out is that the left leg exhibited a significantly higher maximal anteroposterior force in the post-fatigue phase (0.84% ± 0.10) compared to the pre-fatigue phase (0.57% ± 0.24), with a *t*-value of 3.60 and a *p* value of 0.002. However, no significant differences were observed in other force vectors between the pre- and post-fatigue phases.

## 4. Discussion

The subjective rating of perceived exertion (RPE) was used to assess the level of fatigue experienced by the participants after the fatigue task. The RPE score of 17 ± 0.98 indicated that the participants reached a very strenuous level of fatigue. This high RPE score confirms that the athletes experienced significant exhaustion, supporting the effectiveness of the fatigue protocols used in this experiment.

### 4.1. Muscle Activity of Plantarflexion Decreased after Fatigue

The analysis of electromyography (EMG) data for the soleus (SOL) and gastrocnemius lateralis (GL) muscles before and after fatigue revealed several notable findings. Specifically, the maximal muscle activity levels of the SOL and GL in both legs significantly decreased after the fatigue task, with the decline being more pronounced in the right leg. Additionally, the results indicated a reduction in heel height following fatigue, which further supported our first hypothesis that fatigue leads to decreased ankle flexor activity and a subsequent drop in heel height. The decrease in muscle activity after fatigue suggests that, during the later phases of the test, the athletes experienced reduced ankle joint flexion. This reduction in flexion hampered their ability to execute movements required to hit the ball effectively. Previous studies reported that muscle fatigue can have a significant impact on athlete performance [30,31]. When muscles are fatigued, their ability to generate force and sustain optimal levels of activation diminishes [13]. Consequently, this diminished ability compromises movement patterns, coordination, and overall performance [32]. When plantarflexion is limited by fatigue, athletes are less able to push off the ground forcefully, leading to reduced acceleration and deceleration capabilities. This impaired movement affects their ability to reach and strike the ball with precision. Alterations in plantarflexion biomechanics can lead to compensatory movements and increased stress on other joints and structures. Such changes may result in altered movement patterns, greater strain on the knee and hip joints, and a heightened risk of overuse injuries in surrounding muscles and tendons.

Our findings also showed a more significant decline in muscle activity in the right leg compared to the left. Studies [33,34] have shown that the dominant leg typically exhibits better neuromuscular control, greater knee flexion, and less knee valgus and tibial rotation under both pre- and post-fatigue conditions. This increased demand on the dominant leg during physical activities may contribute to the observed differences in muscle activity and fatigue levels between the two legs. This observation suggests that the fatigue-induced impact on muscle performance was uneven between the lower limbs. The muscles of the right leg, including the plantarflexion, may have experienced greater fatigue due to increased involvement during play. Reduced muscle activity in the right leg can potentially impair movement control, balance, and overall performance [35,36]. Asymmetrical muscle fatigue may contribute to imbalances in force distribution and joint stability, which could increase the risk of injury [37,38,39].

Furthermore, previous research has indicated that sports injuries often occur during the latter stages of training or competition [37,40]. Sperlich et al. [41] found that the intensity during table tennis competitions was lower than that of table tennis training, which highlights the importance of recognizing and addressing asymmetry in muscle fatigue. Coaches and athletes must implement strategies to mitigate this risk by targeting the weaker side of the body. Strengthening exercises aimed specifically at underdeveloped muscles can help restore balance and improve overall performance. Additionally, ensuring appropriate training levels and incorporating sufficient rest periods are crucial. Overtraining, inadequate recovery, and insufficient rest exacerbate muscle imbalances and increase the risk of injuries [42,43]. Well-designed training programs that balance intensity, volume, and recovery are essential for reducing the risk of injury and enhancing performance [44].

### 4.2. Co-Contraction Index Increased from Pre to Post Fatigue

The movement of the ankle joint is crucial for lower body actuation and plays a significant role in generating hitting speed in table tennis. Seeley et al. [45] identified a positive correlation between hitting speed and peak angular velocity of the ankle joint. The ankle joint contributes to the kinetic chain and power generation in table tennis strokes by serving as a vital pivot point that facilitates energy transfer from the lower limbs to the upper body, ultimately influencing the final hitting speed [36,46].

Our results revealed an intriguing finding: the maximal anteroposterior ground reaction force (GRF) generated by the left leg increased during the post-fatigue phase compared to the pre-fatigue phase. Interestingly, no significant changes were observed in the medial–lateral and vertical components of GRF between the pre- and post-fatigue phases. This outcome challenges our initial second hypothesis, which suggested that reduced plantarflexion due to fatigue would negatively affect the generation of ground reaction forces (GRFs).

This unexpected result raises questions about the underlying mechanisms responsible for this phenomenon. Compensatory strategies or altered movement patterns [47,48,49] may have contributed to this increase in anteroposterior GRF. During the forward phase of a table tennis stroke, the athlete performs movements that involve kicking the ground with the right foot and rotating the body forward to the left. After striking the ball, the racket arm swings forward while the center of gravity shifts toward the left foot. These movements require precise coordination and balance. Gu et al. [35] found that fatigue significantly decreases the balance of athletes during table tennis multi-ball training, which suggests that fatigue-induced limitations in plantarflexion may disrupt balance and overall performance. To maintain balance and stability during these dynamic movements, athletes may need to exert greater force with their left foot to compensate for reduced plantarflexion. This compensatory force helps preserve coordination and stability, potentially explaining the increased anteroposterior GRF observed in the post-fatigue phase.

Understanding the impact of fatigue on GRF and plantarflexion provides valuable insights for athletes, coaches, and sports scientists. It highlights the importance of managing fatigue levels during training and competition while also emphasizing the need for strategies that enhance ankle flexibility and strength. Addressing these factors can help athletes optimize force production, reduce the risk of injury, and improve overall performance in table tennis.

However, this study had certain limitations. The participant pool was limited to high-level athletes, which may restrict the generalizability of the findings to the broader population of table tennis players. Additionally, while we examined the relationship between ankle flexion and ground reaction forces, we did not directly assess the impact of these changes on specific performance outcomes such as shot accuracy or overall game success. Finally, this study focused solely on acute fatigue induced by a specific protocol, leaving the effects of long-term or chronic fatigue unexplored. Future research should address these limitations by including a more diverse sample of players, examining the direct effects of fatigue on performance outcomes, and investigating the impact of chronic fatigue. This would provide a more comprehensive understanding of fatigue’s role in table tennis and inform the development of effective training and injury prevention strategies.

## 5. Conclusions

This study demonstrated that fatigue significantly impacts muscle activity, ground reaction forces, and ankle flexibility in table tennis players, particularly highlighting the presence of asymmetrical muscle fatigue in the lower limbs. The decrease in muscle activity, especially in the right ankle flexors, leads to reduced joint flexibility, while the compensatory increase in anteroposterior ground reaction force in the left leg suggests adaptive strategies to maintain performance. These findings underscore the importance of managing fatigue levels, addressing muscle imbalances, and improving ankle flexibility and strength to optimize performance and reduce the risk of injuries in table tennis.

## Figures and Tables

**Figure 1 sensors-24-06521-f001:**
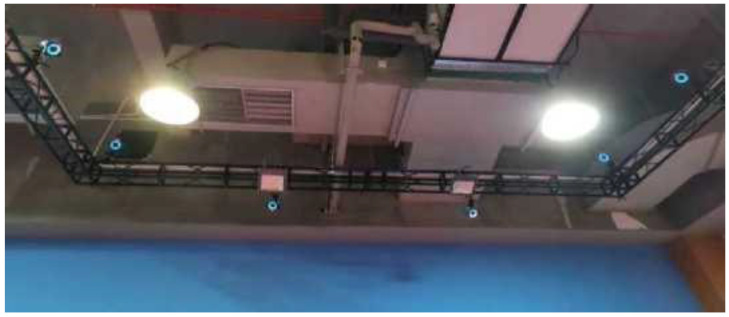
Motion capture system.

**Figure 2 sensors-24-06521-f002:**
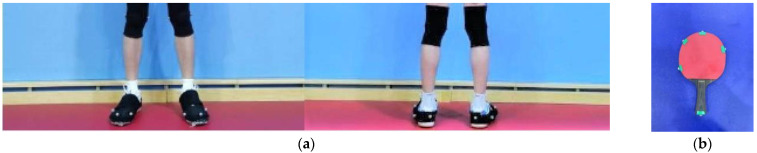
Motion capture system and markers’ positions on the foot (**a**) and the racket (**b**).

**Figure 3 sensors-24-06521-f003:**
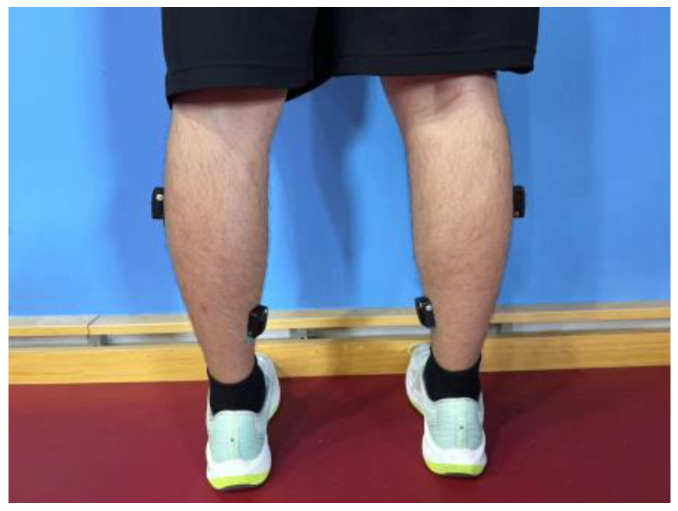
Electrode positions on both legs.

**Figure 4 sensors-24-06521-f004:**
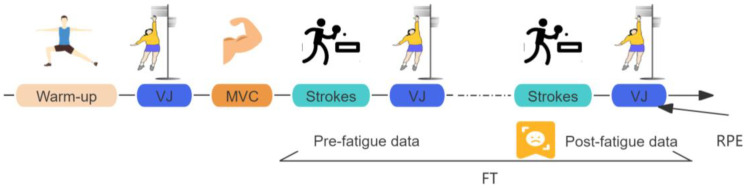
Overview of the experimental protocol. VJ = vertical jump test; MH = maximal vertical jump height; FT = fatigue task; strokes included 20 forehand strokes. FT was continued until the vertical jump height was lower than 70% of the MH.

**Figure 5 sensors-24-06521-f005:**
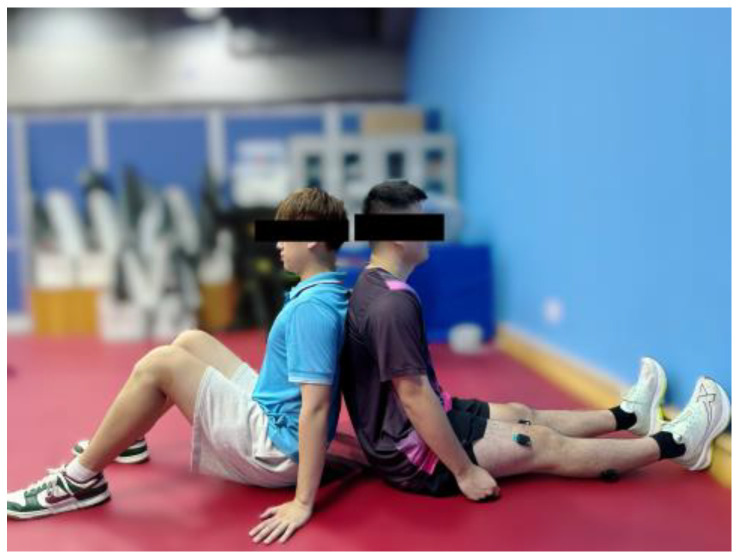
Maximal voluntary contraction (MVC) test.

**Figure 6 sensors-24-06521-f006:**
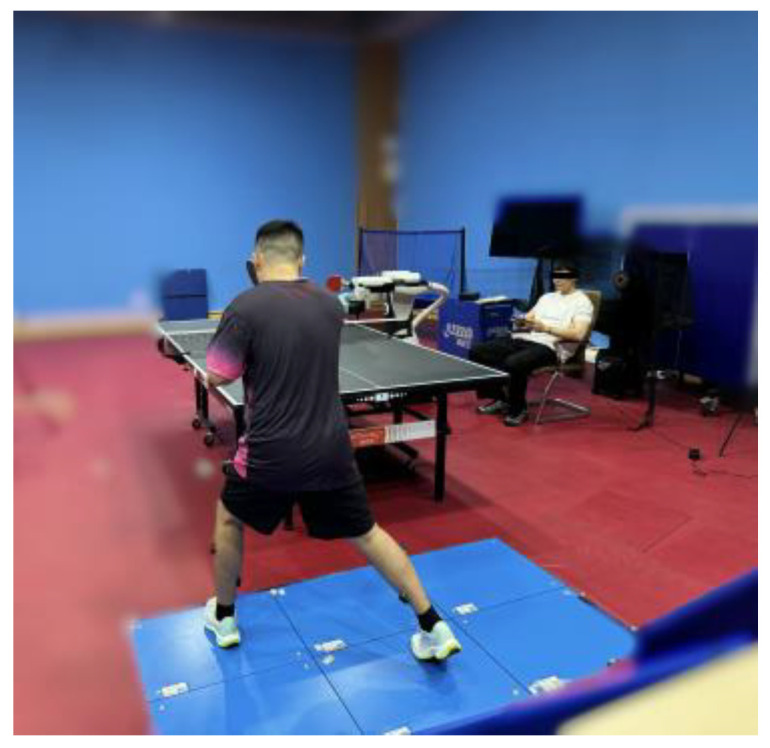
Experimental environment and fatigue task.

**Figure 7 sensors-24-06521-f007:**
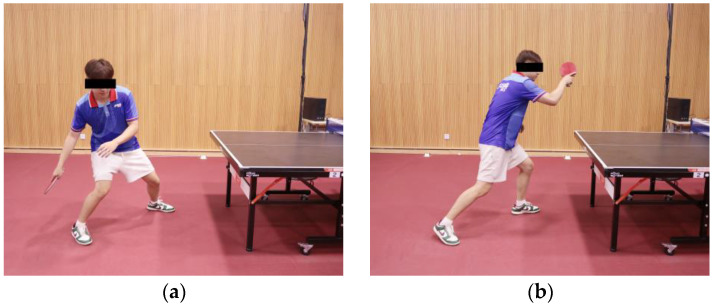
Division and definition of motion phase [27]: (**a**,**b**) was defined as the forward phase (FS); position (**a**) was defined as the key event that indicated the end of the BS; position (**b**) was defined as the key event that indicated the end of the FS.

**Figure 8 sensors-24-06521-f008:**
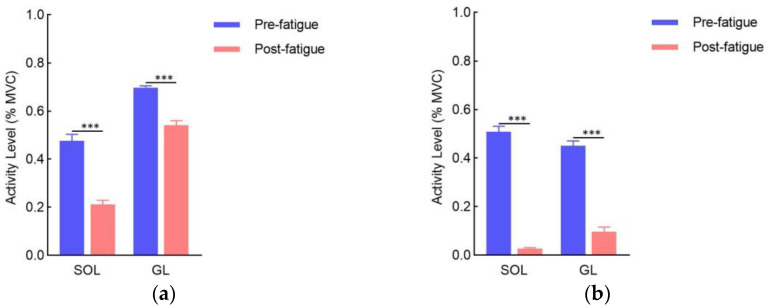
Muscle activity level during the pre- and post-fatigue phases: (**a**) left leg; (**b**) right leg. SOL: soleus, GL: gastrocnemius lateralis; EMG signals were normalized to the muscle activity assessed during the maximal voluntary contraction (MVC); *** indicates significant differences between pre- and post-fatigue phases (*p* < 0.001).

**Figure 9 sensors-24-06521-f009:**
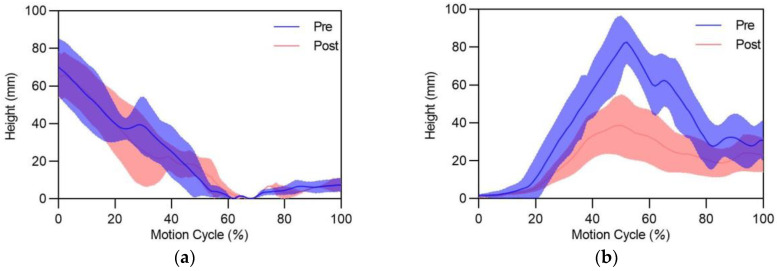
The trend of heel height during the pre- and post-fatigue tasks: (**a**) left foot; (**b**) right foot.

**Table 1 sensors-24-06521-t001:** Maximal ground reaction force (GRF) during the pre- and post-fatigue phases.

Leg	GRF Vectors	Pre-Fatigue	Post-Fatigue	*p* Value
Left	*F_x_*	0.57 ± 0.24	0.84 ± 0.10	0.002 *
	*F_y_*	1.69 ± 1.18	1.39 ± 0.92	0.495
	*F_z_*	81.45 ± 0.19	81.17 ± 0.98	0.340
Right	*F_x_*	1.21 ± 0.76	1.15 ± 0.73	0.845
	*F_y_*	4.86 ± 0.81	4.59 ± 1.02	0.480
	*F_z_*	124.23 ± 2.89	124.10 ± 3.50	0.976

Notes: All forces (*N*) were normalized by dividing them by the body weight; the unit is %; *F_x_* = anteroposterior GRF, *F_y_* = mediolateral GRF, *F_z_* = vertical GRF. * indicates significant differences between pre- and post-fatigue phases (*p* < 0.01).

## Data Availability

The data that supports the findings of this study are available from the corresponding authors upon reasonable request.

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
