# Peer review of "Effects of Fatigue on Ankle Flexor Activity and Ground Reaction Forces in Elite Table Tennis Players"

_sensors, 2024, doi:10.3390/s24206521_

Round 1

Reviewer 1 Report

Comments and Suggestions for Authors

General: The current manuscript looks to evaluate the effects of fatigue on ankle flexion and ground reaction force in elite table tennis players. The authors found a significant increase in the level of fatigue in the right leg, which affected its ability to lift the heel off the ground. While the manuscript is well written, I believe there are some changes needed to improve the overall manuscript.

COMMENTS

General: For the most part, the manuscript uses the term ankle flexors or ankle flexion. However, at times you interchange this with plantarflexion. For consistency in the manuscript you should pick one term and stick with it, even though both are correct.

Abstract

Page 1, Line 20: Your statement about “potentially impairing ball-striking ability.” should be removed. Your study didn’t examine ball-striking ability.

Introduction

Page 1, Lines 36-37: Your sentence starting with “Notably, they are frequently…” seems out of place in the context of the paragraph and should be removed or you need to expand on the connection between the upper and lower body in table tennis.

Page 2, Lines 62-63: Your sentence starting with “Several studies have….” Should be removed as you already stated mental and physical fatigue affect performance above.

Page 2, Line 63: Change the word “Specifically” to “On the other hand”. As you are switching between muscle to mental fatigue, it makes for a more clear transition.

Methods

Figure 4: I was really confused with this figure on what was happening between pre-fatigue and post-fatigue. You need to find someway to clear up that multiple sets occur depending on fatigue.

Figure 4: You have a typo in “Pre-fatigue data”. It currently reads ‘pro’.

Page 5, Lines 179-190: For your MVC test, you need to be clear on whether this was an isometric or concentric contraction.

Page 5, Line 195: You need to make sure you have complete sentences. It should read something like, “The fatigue task in this study followed protocols performed in previous research. [23, 24]”

Page 6, Lines 202-204: You’ll need to clarify which Borg RPE scale was used, 1-10 or 6-20.

Page 6, Lines 219-220: Again, you’ll need to complete this sentence.

Results

You need to include the average number of sets until fatigue along with a range.

Discussion

Page 9, Lines 310-311: This statement needs a reference.

Page 9, Lines 311-212: This statement needs a reference.

Page 9, Lines 321-328: Can you explain why the fatigue was greater in the right leg than the leg?

Page 9, Lines 336-338: This sentence needs a reference.

Page 10, Line 350: Medial should be medial-lateral as movement can happen in both directions.

Conclusion

Page 10, Lines 377-378: You didn’t measure ball striking, so you can’t state this as a conclusion to your study.

Page 10, Lines 382-393: Your limitations and future research should be at the end of your discussion. The conclusion is just revisiting the highlights of your study and how it affects the real world.

Reviewer 2 Report

Comments and Suggestions for Authors

In this manuscript, the fatigue effects on ankle flexor activity and ground reaction forces in elite table tennis players were investigated. Muscle activity of the soleus and gastrocnemius lateralis muscles, heel height, and GRFs were measured using a combination of wireless electromyography, motion capture, and force plate systems. This work is meaningful for sensors application and the compensatory strategies to maintain balance and performance were proved effective. I would like to recommend the paper for publication in sensors.

Minor suggestion:

The analysis of differences between athletes is lacking.

The possible application range of this evaluation method can be prospected.

Comments on the Quality of English Language

good
